# Population Genetics Revealed a New Locus That Underwent Positive Selection in Barley

**DOI:** 10.3390/ijms20010202

**Published:** 2019-01-08

**Authors:** Stephan Reinert, Alina Osthoff, Jens Léon, Ali Ahmad Naz

**Affiliations:** Institute of Crop Science and Resource Conservation, Plant Breeding, University of Bonn, Katzenburgweg 5, 53115 Bonn, Germany; Stephan.reinert@colorado.edu (S.R.); s7alosth@uni-bonn.de (A.O.); j.leon@uni-bonn.de (J.L.)

**Keywords:** population genomics, outlier loci, positive selection, barley, phylolochron

## Abstract

Trait variation among natural populations and their cultivated relatives occurs due to evolutionary forces, including selection and drift. In the present study, we analyzed these forces at the locus level in a global barley diversity set using population genetics analysis. Genome-wide outlier loci detection found a locus on chromosome 2H at which a common single nucleotide polymorphism (SNP) marker SCRI_RS_170235 accounted for the highest diversity index (Fst) values between cultivars and landraces and between cultivars and wild accessions. For a population wide genetic analysis, we developed a Polymerase Chain Reaction (PCR)-based cleaved amplified polymorphic marker at the identified locus. Marker genotyping of 115 genotypes identified a characteristic distribution of polymorphisms among the cultivated, landraces, and wild barley accessions. Using this marker, we screened a library of wild barley introgression lines (IL) and selected IL S42IL-109 that carried the wild introgression of the outlier locus in cultivar ‘Scarlett’ background. A plethora of phenotypic evaluation was performed between the S42IL109 and ‘Scarlett’ to dissect the putative effect of the identified outlier locus. Comparison of S42IL109 and ‘Scarlett’ revealed significant difference in the development of phyllochron two (Phyl-2), phyllochron three (Phyl-3), and phyllochron four (Phyl-4). Across the three phyllochrons, it was consistently observed that S42IL109 developed successive leaves in a shorter time span, by one to two days, compared to ‘Scarlett’. These data suggest that outlier locus may influence phyllochron variation which underwent positive selection in barley.

## 1. Introduction

Barley is one of the most diverse crop species that belongs to the largest tribe of monocotyledonous plants, Triticeae of family Poaceae. The genus Hordeum comprise of around 32 species and 45 taxa, including annual and perennial species. Once a foundation crop of Old World agriculture, barley is now being grown extensively from boreal to equatorial regions across the world [1]. Its vast diversification and ecological adaptation is the result of evolutionary forces like mutation, natural selection, drift, and migration. The effect of these forces is visible in the form of trait diversity and population differentiation among its natural populations of cultivars, landraces, and wild accessions. Such wide population diversification among the natural genetic resources adapted to different climatic conditions offers an opportunity to explore primary steps of evolution and domestication in barley.

The cultivated barley (*Hordeum vulgare* ssp. *vulgare*) originated from wild barley (*Hordeum vulgare* ssp. *spontaneum*) through the process of domestication. Remains found in the Middle East suggests that domestication occurred around 8000 BCE in an area referred as the Fertile Crescent [2,3]. Wild accessions are still colonized in the area of the Fertile Crescent [4]. Multiple genome studies have revealed the Fertile Crescent as the origin of barley domestication [5,6,7]. Several studies have discovered wild barley in Greece, Ethiopia, Egypt, and Asia in addition to the Fertile Crescent [3,8]. These reports show an interesting insight on the origin of barley and how it was able to populate nearly the entire globe. Increasing evidence support the theory of multiple sites of barley domestication in Asia in addition to the Fertile Crescent [9,10,11].

Genetic determination of evolutionary footprints of natural selection and domestication in cereals is made by using different strategies based on population genetics and genome analyses. A “forward genetics” approach aims to identify the genes based on a specific trait phenotype using marker by trait association analyses. In contrast, a “reverse genetics” approach focuses on analyzing the function of a particular gene by studying the effects of sequence changes on the phenotypic trait. Both approaches led to the discovery of several domestication-related genes and mutations [12,13]. A major change from wild to domesticated forms was the reduction in grain shattering to facilitate harvest. A shattering QTL in rice, called *Sh4*, explains ~70% of phenotypic differences between a rice cultivar and its wild ancestor, *Oryza nivara*. The substitution of a single nucleotide led to a mutation in a putative transcription factor, changing the amino acid from lysine to asparagine. This mutation caused a weakening in gene function and therefore inhibits the shattering of grains upon ripening [14]. In wheat, a major domestication gene, *Q*, was identified and found to affect the formation of a non-brittle rachis, shorter spikes, and shorter culms [15]. Similarly, Avni et al. used modern genomic techniques to identify and analyze genetic modifications which underlay domestication of wild wheat. They identified a genomic region with signatures of a domestication-related selective pressure which revealed the causal mutation of *Brittle Rachis 1* (*TtBtr1*) genes controlling shattering in Poaceae [16]. In barley two closely linked genes, *non-brittle rachis1* (*Btr1*) and *non-brittle rachis2* (*Btr2*), control the fixation of non-brittle rachis [17]. Similarly, the occurrence of six-rowed forms in barley was highly important for a positive selection resulting in higher yield. The gene controlling the two-row phenotype, *Vrs1*, encodes a transcription factor that is expressed in the lateral-spikelets and leads to the formation of infertile spikelets. A mutation leading to a loss of function of this transcription factor enables plants to develop fertile lateral-spikelets and six-row ears [18]. Further, Akpinar et al. identified differences in content and distribution of repetitive sequences between the subgenomes A and B in wild emmer wheat which lead to small-scale evolutionary rearrangements as well as provided evidence for the 4AL-5AL-7BS translocation [19]. Hence, *Btr1*, *Btr2*, *Vrs1*, *Nud*, and *Bkn-3* are considered the primary domestication-related genes in barley [5,7,17,18,20]. Recently, the “reverse genetics” approach is becoming the method of choice for evolution and population genetics studies due to the current revolution in genomics. More notable genome sequencing and high-throughput genotyping (array or genotyping by sequencing) allows a high resolution dissection of evolutionary events across the genome in term of single nucleotide polymorphism (SNP) markers [10,20]. In barley, Fang et al. studied 318 genotypes of barley originating from different geographic regions using SNP array genotyping and predicted two genomics regions on chromosome 2H and 5H, which were distributed disproportionately to geographic differentiation [21]. Abebe et al. performed landscape genomics of diverse Ethiopian landraces using genotyping by sequencing and identified adaptive loci which are affected by selection. In total, 80 putative adaptive loci were detected among the barley landraces by analyzing climate variables and geographic distances [22].

The primary objective of the present research was to study genome-wide allele dynamics in a global barley diversity set comprising of cultivars, landraces, and wild barley accessions to scan loci influenced by natural or artificial selection. Population genetics analyses of these resources identified a new locus on chromosome 2H that underwent positive selection in barley.

## 2. Results

### 2.1. Population Structure Analysis

Population structure was calculated to see population differentiation across the barley diversity set. The best K-value detection implemented in Clustering Markov Packager Across K (CLUMPAK) revealed three distinct subclusters (SPOPs) within the barley population (Appendix A). Based on the membership coefficient (MCo), a total of three SPOPs as well as an admixture group (ADMIX) have been identified. Only 115 out of the 179 accessions within the barley population were grouped into the three SPOPs, 64 accessions were grouped into ADMIX due to a MCo < 0.85 (Figure 1). A detailed analysis of the three SPOPs revealed a subspecies specific distribution of barley accessions with SPOP 1 containing 35 wild forms and 13 landraces; SPOP 2 consisting of 13 cultivars, 27 landarces, and 2 wild forms; and SPOP 3 being made of 22 cultivars and 3 landraces. Moreover, the analysis revealed a SPOP specific geographic distribution (confidence interval ≥ 75%). Based on the site of collecting SPOP 1 is considered as the Middle Eastern/Asian cluster, SPOP 2 is the American/European cluster, and SPOP 3 represented a European cluster (Figure 1).

We calculated the total genetic distribution at the genome level for a haplotype analysis and compared the genetic background of these haplotypes. A close genetic relatedness of genotypes within and among SPOPs has been identified (Appendix A).

### 2.2. Genomic Scan for Outlier Loci Detection

In order to identify evidence of selection associated with polymorphic SNP markers we used the loci outlier tool BayeScan. A total of 66 significant outlier loci have been observed within the entire population (Figure 2A). We detected four outlier loci between SPOPs 1 and 3 as well as one locus between SPOP 1 and SPOP 2 (Figure 2B,C). The detected loci revealed positive alpha values which is an indicator of directional selection. Interestingly, one out of the five significant outlier loci were common in the comparisons of SPOP 1 and SPOP 2 as well as SPOP 1 and SPOP 3 (Figure 2D). The one significant locus for the SPOP 1/SPOP 2 comparison showed the highest F_ST_ value (F_ST_ = 0.41) among all identified outlier. The F_ST_ values for the rest of the outliers, identified between SPOP 1 and SPOP 3, ranged from 0.27 to 0.37. One detected outlier locus was common among the comparisons SPOP 1/SPOP 2 and SPOP 1/SPOP 3 and could be identified as SNP marker SCRI_RS_170235 located on chromosome 2H. The identified SNP marker SCRI_RS_170235 has been used to identify the region for candidate gene detection as well as to develop the CASP marker. Fst distribution of entire population ranged from 0.27 to 0.75 (Appendix A).

### 2.3. Population Genetic Analysis

The most significant SNP marker SCRI_RS_170235 was positioned in the gene AK366024 annotated as predicted protein (Protein ID: BAJ97227.1) in barley. We performed a full-length sequencing of this gene in the selected pool of cultivars and wild barley accessions. Sequencing comparison revealed that all wild barley accessions carried the nucleotide thymine (T) at position 224 bp from the start codon which was deleted in all selected cultivars. Next, we established a deletion specific cleaved amplified polymorphic sequence (CAPS) marker for population-wide genotyping of this locus specific marker in the barley diversity set. After restriction digest using AvaII, genotypes carrying the T insertion exhibited two restricted fragments (75 bp + 227 bp). Genotypes carrying the T deletion showed one large uncut PCR fragment of 302 bp. Based on CAPS genotyping, we evaluated the distribution of this mutation across the diversity panel. It revealed that 100% of the genotypes within SPOP 3 (mostly cultivars) inherited the T deletion and SPOP 1 (mostly wild barley) possessed the T. In SPOP 2, 52.38% of genotypes showed T deletion, whereas 38.10% genotypes harbored the T. Moreover; 9.52% of the genotypes within SPOP 2 were heterozygous (Appendix A).

### 2.4. Phenotypic Evaluation of Outlier Locus Alleles Carrying Genotypes

To test the putative role of the common outlier locus, we identified an introgression line (S42IL109) that carried wild barley introgression at the outlier locus in the ‘Scarlett’ background using deletion-based gene specific marker (Appendix A). Initially, a plethora of phenotypic evaluations were performed in S42IL109 and the recurrent parent ‘Scarlett’ under controlled conditions at seedling stage. The comparison of trait variation found significant differences in the development of phyllochron two (Phyl-2), phyllochron three (Phyl-3), and phyllochron four (Phyl-4) between ‘Scarlett’ and S42IL109. Across the three phyllochrons, it was consistently observed that S42IL109 developed successive leaves in a shorter time span, by one to two days, compared to ‘Scarlett’ (Figure 3A,B and Appendix A). Phyllochron is measured as intervening time period between the sequential emergence of leaves on the main stem. The development of Phyl-1, the days to germination (DTG), as well as the comparison of seedling and leaf length were nonsignificant between ‘Scarlett’ and S42IL109.

An additional experiment was performed to evaluate phenotypic differences in late plant development as well as to see the effects of drought in ‘Scarlett’ and S42IL109. This analysis detected significant differences between ‘Scarlett’ and S42IL109 for the BBCH growth stages under controlled and drought conditions. Under controlled conditions, ‘Scarlett’ showed a lower mean BBCH stage (56) compared to S42IL109, with a mean BBCH stage of 60 at harvest. Similarly, under drought conditions, ‘Scarlett’ showed a lower mean BBCH stage of 58 compared to a BBCH stage of 61 in S42IL109 (Figure 3C and Figure 4). The phyllochron development and overall mean comparison of BBCH development stages between ‘Scarlett’ and S42IL109 showed a delayed growth and development in ‘Scarlett’ in comparison to S42IL109.

## 3. Discussion

Directional selection during domestication has played a fundamental role to produce genetic divergence between cultivated varieties and their wild ancestors. In this process, some of the useful alleles/traits underwent positive selection, which became prevalent during intensive breeding among the cultivated gene pool [23,24,25,26]. The primary focus of this study was to explore a diverse barley population with accessions adapted to different climates and developed under different selection pressures, to observe allele frequency shifts as they underwent natural or artificial selections in domestication. We followed a population genetic analysis using a Bayesian-based method to detect loci affected by selection. The three SPOPs were compared against each other revealing a total of five outlier loci. Four outlier loci were detected between SPOP 1 (wild barley) and SPOP 3 (cultivars), one outlier locus was detected between SPOP 1 and SPOP 2, but no outlier locus between SPOP 2 and SPOP 3 was detected. A higher number of outlier loci were detected between the SPOP 1 and SPOP 3, which was expected due to divergence and the protracted evolution history of both subpopulations. These results are in line with the geographical distribution of the selected genotypes, where wild barley population revealed high genetic diversity compared to modern cultivars due to morphological and physiological adaptations [27]. Notably, the most significant outlier locus was common between SPOP 1 and SPOP 3 and between SPOP 1 and SPOP 2, but this locus was not detected between SPOP 2 and SPOP 3, suggesting a selection bottleneck occurred which resulted in the allele frequency shift of this locus during domestication. Hübner et al. discovered structural distribution of barley landraces in Sardinia compared to other landraces and modern cultivars [26]. Similarly, Allaby (2015) determined that the high degree of adaptation in barley occurs because of several centers of origin in the Middle East and Asia which is supported by the results of Morrell et al. (2007) and Dai et al. (2012), as well as with the geographical distribution of wild barley presented in the present study [10,11,28].

Next, we evaluated the putative effect of identified outlier locus due to trait variation. For this we identified an introgression line (IL), S42IL109 carrying wild segment from wild barley at the outlier locus in the cultivated barley background of ‘Scarlett’. For phenotyping, quantitative traits ILs provide a major advantage: not all but most epistatic effects are eliminated so every variation in the phenotype between the elite cultivar and the IL is directly associated with the introgressed segment, which increases the possibility to identify even small phenotypic effects [29]. The applicability of introgression lines was tested in several studies in a broad variety of species like melon [30], tomato [31], and wheat [32]. The comparison of S42IL109 and ‘Scarlett’ resulted in an opportunity to compare both alleles and their roles in the determination of traits. We performed several phenotypic evaluations between S42IL109 and ‘Scarlett’, but the most promising differences were detected in the phyllochrons. S42IL109 is an introgression line which carries wild barley introgression at the outlier locus on 2H, this introgression harbors the wild allele of AK366024 as compared to ‘Scarlett’. It is possible that the variation in AK366024 may be associated with observed phyllochron differences between S42IL109 and ‘Scarlett’. However, there are additional candidate genes on the wild introgressions which require further research. A phyllochron is the intervening time period between the sequential emergence of leaves on the main stem, and therefore considered very useful to evaluate plant growth and development [33]. Leaf emergence is a genetic trait which is influenced by environmental cues like temperature, rainfall and nutrients availability. A precise determination of phyllochrons was important to dissect adaptive evolution in general as well as the microevolution of quantitative traits that underwent directional selection in crop plants like barley. Previous studies on domestication identified several genes, such as *Bkn-3*, *Vrs1*, *Nud*, and *Btr1/Btr2* that revealed major phenotypic effects [5,7,15,16,17,18].

In summary, the developmental differences, described in the form of a pictogram, suggest that there were no significant differences between ‘Scarlett’ and S42IL109 until Phyl-2 eight days after sowing: there was slower development in ‘Scarlett’ and faster growth in S42IL109. The slower growth and development in ‘Scarlett’ was also visible at later development stages, such as BBCH 22 to 60, which leads to early heading and maturity in S42IL109. Based on this variation, we propose that the faster growth, development, and maturity in S42IL109, could be associated with an imbalance of water and nutrients uptake as compared to cultivar ‘Scarlett’. We hypothesize that a relatively delayed development and growth in ‘Scarlett’ might be advantageous in the establishment of more tillers, biomass and longer grain filling, that might have underwent positive selection in cultivated barley.

## 4. Materials and Methods

### 4.1. Plant Material

A total of 179 barley genotypes collected across 38 countries were analyzed in the present study. The population includes a variety of *Hordeum vulgare* ssp. *spontaneum* (48), and *Hordeum vulgare* L. ssp. *vulgare* (131) accessions. The 131 *Hordeum vulgare* L. ssp. *vulgare* accessions were composed of 72 landraces and 59 modern cultivars [34] (Appendix A). The Leibniz Institute for Plant Genetic and Crop Science (IPK, Gartersleben, Germany), Nordgen (NGB, Alnarp, Sweden) and the International Center for Agricultural Research in the Dry Areas (ICARDA, Beirut, Lebanon) provided the seeds for this study.

To study the phenotypic effect of significant outlier loci, a wild barley introgression library comprising of 72 lines was used. From these 72 lines, several introgression lines carry multiple introgression with the exception of line S42IL109. A cross between the wild Israeli barley accession ISR42-8 and the German spring barley cultivar ‘Scarlett’ provided the initial step for the barley introgression lines population (S42ILs) analyzed in the present study. Further details about the breeding history of the IL population can be found in Schmalenbach et al. (2008) [35].

The 179 barley genotypes have been separated into three groups (SPOP) and one admixture group (ADMIX) based on their genetic relatedness defined by STRUCTURE. Genotypes where grouped into one of the three SPOPs when their membership coefficient (MCo) was 0.85 or higher for one of the three groups. Based on that, 35 wild accessions and 13 landraces were grouped in SPOP1, 13 modern cultivars, 27 landraces, and three wild accessions were grouped together in SPOP2, and 22 modern cultivars as well as three landraces were grouped in SPOP3. The exact composition of each group can be found in Appendix A.

#### Genotyping

Genotyping has been performed by using the Illumina 9K iSelect SNP chip, at TraitGenetics (TraitGenetics GmbH, Seeland OT Gatersleben, Germany) [36]. A total of 7842 SNP markers were obtained and filtered as described by Miyagawa et al. (2008) [37]. The minor allele frequency (MAF) was set to >0.05, with an SNP call rate of >0.95, and missing values <0.05. After removing all monomorphic markers, using SAS 9.3 (SAS Institute 2008, Cary, NC, USA), a total of 5892 markers remained and were used in the following analyses.

### 4.2. Population Structure Analysis

We analyzed the population structure using the software package STRUCTURE v2.3.4 (Pritchard Lab, Stanford University, Stanford, CA, USA). Software settings were set to a default admixture and independent allele frequency models, with 1 to 20 K-values, 100,000 burnin period length, 300,000 MCMC replications, and 10 iterations (according to Morrell and Clegg (2007) [11]). We performed the detection of the best ∆K using a Markov clustering algorithm implemented in the software package CLUMPAK [38]. The detection is based on the method of Evanno et al. [39] and ln(Pr(X|K) values (STRUCTURE’s manual, Section 4.1). The method of Evanno enables the detection of a K-value out of a range of K-values which identifies the uppermost level of structure. The ln(Pr(X|K) values reveal k for which Pr(K = k) is highest. A genetic distances matrix was constructed based on the set of 5892 polymorphic SNP markers. The matrix was used to estimate a neighbor-joining phylogenetic tree with 1000 bootstraps (DARwin 6, Paris, France) [40].

### 4.3. Loci Outlier Analysis

We used the SNP outlier tool BayeScan 2.1 (Matthieu Foll, University of Bern, Bern, Switzerland) to identify SNP outlier among different SPOPs. BayeScan is based on the multinomial-Dirichlet model and detects loci which show a higher or lower level of divergence compared to neutral loci. Posterior odds (PO) is the ratio of the posterior probability, based on the allele frequency, of the two models ‘selection/neutral’ and shows the probability of selection of a given locus. We assumed that the neutral model is 10 times more likely than the model including selection. Default settings were kept with 10 pilot runs each 5000 iterations and 50,000 burn-in steps. The significance of detection was determined using false discovery rate (FDR ≤ 0.05) which was controlled by the q-value. A membership coefficient (MCo) has been calculated to group accession into different subgroups (SPOP). The MCo was set to 0.85. Genotypes with less than 0.85 genetic identity of one of the three subgroups were considered as admixture and removed from the population.

The diversity index (Fst) statistics have been calculated using the Fst function from the hierfstat package in R based on Weir and Hill [41].

### 4.4. Restriction Fragment Polymorphism Analysis

To analyze the distribution of identified outlier loci, we used the Cleaved Amplified Polymorphic Sequence (CAPS) genotyping approach [35]. The CAPS marker, which we developed, derived from a specific deletion which was specific to a significant SNP marker. We identified a unique restriction sit close to the marker by using DNAStar SeqBuilder (DNASTAR^®^ Inc., Madison, WI, USA). To amplify the 302 bp DNA fragment, a specific pair of PCR primer was designed (Appendix A, CAPS). A restriction digest with the restriction endonuclease AvaII, results in two fragments with a length of 227 bp and 75 bp, when the restriction site was present in the amplicon.

### 4.5. Phenotypic Comparison of Barley Plant Development

The experiments for the phenotypic evaluation were performed inside a greenhouse for whole plant development, as well as in a climate chamber (Viessmann Kältetechnik AG, Hof/Saale, Germany) for seedling and phytomer development. The climate chamber setup enabled controlled conditions, which is crucial for determination of phyllochron.

Barley development can be divided into several stages from germination to maturity. A commonly used scale to measure growth and development of cereal crops is called BBCH (Biologische Bundesanstalt, Bundessortenamt und CHemische Industrie) scale [42]. This scale separates the growth and development into ten major stages, spanning from germination (stage 0) to senescence (stage 9), with each major stage consisting of 10 minor stages. According to this, the first leaf of barley emerges is in minor stage 10 of major stage 1 (leaf development). After the emergence of the barley plant’s first leaf, recurring plant structural units, called phytomers, are developed that are characterized by the formation of successive leaves [43]. The intervening time period between the sequential emergences of leaves on the main stem is termed as a phyllochron, which is an important tool to evaluate the growth and development of plants, especially in cereals [44].

### 4.6. Greenhouse Experiment

For comparison of whole plant development based on BBCH stages, 20 plants of ‘Scarlett’, and the introgression line S42IL109 were grown in 5.5 L pots with 40% top soil and 60% silica sand (Cordel & Sohn, Salm, Germany). Temperature was set to 20 °C during the day and 15 °C at night. Until BBCH stages 29–31 were reached, all pots were irrigated automatically using a drip water irrigation system (Netafilm, Adelaide, Australia) three times daily, then half of the plants of each genotype were stressed by completely limiting their water supply for 10 days until visible wilting occurred before returning to regular irrigation. The level of irrigation was determined using the volumetric moisture content (VMC) detected by the DL2e Data Logger soil moisture sensor. Fertilizer and pesticide treatments were applied according to good agricultural practice. The BBCH stage of each plant was measured five times during growth, starting one month after sowing and repeated biweekly until harvest. For statistical analysis, group mean-BBCHs for ‘Scarlett’ and S42IL109 were calculated for each time point. A Student’s *t*-test was applied for pairwise comparison of means to detect significant developmental differences between the genotypes.

### 4.7. Climate Chamber Experiment

To determine differences in seedling growth we grew 40 seeds of ‘Scarlett’, and S42IL109 under controlled conditions in a climate chamber (Viessmann Kältetechnik AG, Hof bei Saale, Germany). A stratification protocol was applied for 48h at 8 °C. After stratification seeds were transplanted into 96-well plates (one seed per well) on compound soil (40% top soil and 60% silica sand) (Cordel & Sohn, Salm, Germany). A 16-h day and 8-h night cycle with ambient temperature of 24 °C (day) and 18 °C (night) was used. Irrigation was applied every other day. After the emergence of the first leaf tip, the phyllochron was measured for successive leaf appearance according to [44]. One week after germination, seedling length as well as length of the first leaf was determined. This measurement was repeated for the second leaf two weeks after germination. To test the differences of development based on phyllochron, mean values for each genotype and phyllochron were calculated and tested for significance based on a pairwise Student’s *t*-test. The same procedure was applied for the comparison of seedling and leaf length.

## Figures and Tables

**Figure 1 ijms-20-00202-f001:**
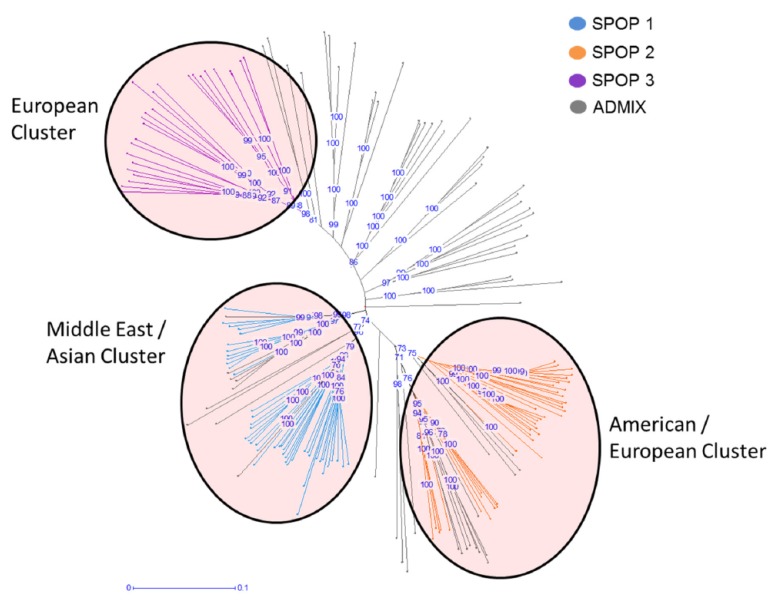
Phylogenetic analysis of barley diversity set. Bootstraps were calculated with 1000 iterations. The blue numbers represent the confidence value in percent. Groups of accessions based on MCo ≥ 0.85. Red transparent circles show the geographic distribution. Blue: SPOP 1 Middle East/Asian Cluster (mostly wild barley). Green: SPOP 2 American/European Cluster (landraces/cultivars). Orange: SPOP 3 European Cluster (mostly cultivars). Gray: ADMIX.

**Figure 2 ijms-20-00202-f002:**
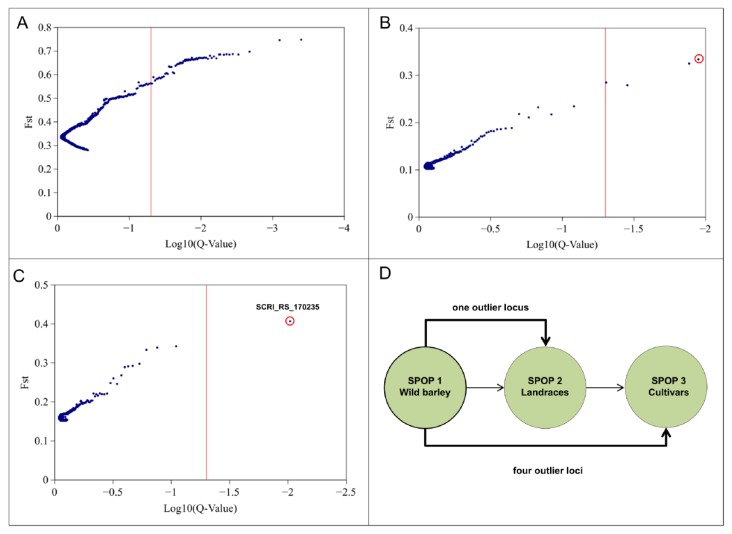
SNP outlier analysis in barley diversity set using Bayescan. (**A**) Outlier locus analysis for the entire population. (**B**) Outlier locus analysis between SPOP 1 and 3. (**C**) Outlier locus analysis between SPOP 1 and 2. Red vertical line indicates threshold (False discovery rate (FDR) ≤ 0.05). Markers on the right site of the threshold are significant outlier. (**D**) Comparison of significant outlier loci among the SPOP 1, 2, and 3. Circle indicate the most significant SNP marker SCRI_RS_170235.

**Figure 3 ijms-20-00202-f003:**
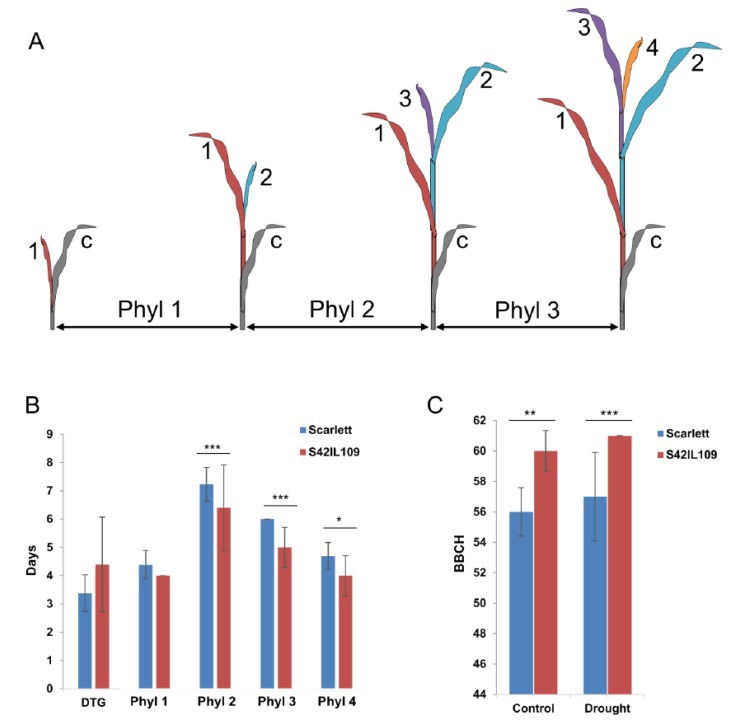
(**A**) Pictogram of phyllochron measurement. Coytledon (c) is depicted in gray; following leaves have ascending numbers (one to four). Tissues with the same color form one recurrent unit called a phytomer. Phyllochron (Phyl) is the time interval between the emergence of leaf tips of successive leaves. (**B**) Mean comparison of developmental traits for genotypes ‘Scarlett’ and S42IL109. Bars represent time in days to germination (DTG), phyllochron 1 (Phyl-1), phyllochron 2 (Phyl-2), phyllochron 3 (Phyl-3), and phyllochron 4 (Phyl-4). (**C**) Mean comparison of trait Biologische Bundesanstalt, Bundessortenamt und CHemische Industrie (BBCH) under control and drought conditions between S42IL109 (red) and ‘Scarlett’ (blue). Asterisks indicate the level of significance at 0.05 (*), 0.01 (**), and 0.001 (***) based on Student’s *t*-tests. Error bars = standard deviation (STD).

**Figure 4 ijms-20-00202-f004:**
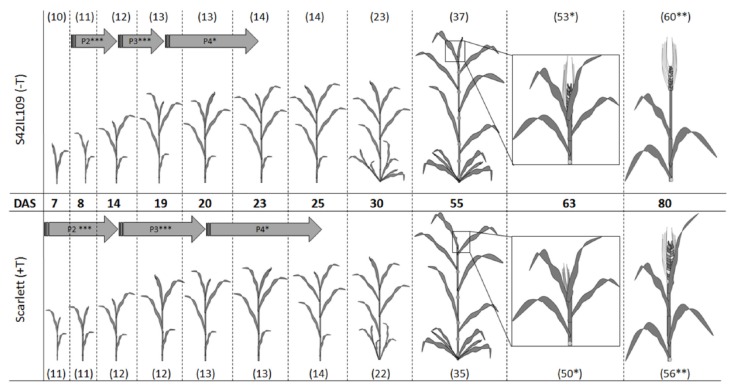
Comparison of developmental traits phyllochron and Biologische Bundesanstalt, Bundessortenamt und CHemische Industrie (BBCH) stages between ‘Scarlett’ and S42IL109 based on days after sowing (DAS). Duration of phyllochrons (Phyl-2 to Phyl-4) is indicated as gray arrows, BBCH growth stage (in numbers), shown in brackets on top and bottom, reveals BBCH stage differences between ‘Scarlett’ and S42IL109. Asterisks show level of significance (* = 0.05, ** = 0.01, and *** = 0.001) based on Student’s *t*-tests.

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
