# Peer review of "Population Genetics Revealed a New Locus That Underwent Positive Selection in Barley"

_ijms, 2019, doi:10.3390/ijms20010202_

Reviewer 1 Report

n this manuscript, a SNP marker on 2H that accounted the highest diversity index (Fst) values between cultivars and landraces and between 17 cultivars and wild accessions was found. The results obtained using this marker suggest that outlier locus may influence phyllochron variation which underwent positive selection in barley. This manuscript is interesting. However, some minor revision is needed.

1. In the Result section, SPOP1, SPOP2, SPOP3 and SPOPs were mentioned. They should be introduced in detail in Materials scetion.

2. The manuscript has detected the difference of the gene AK366024 between wild and cultivar barley. Whether did this gene of wild barley exists in the introgression line S42IL109? Were the trait variations between line S42IL109 and Scarlett caused by the SNP in this gene? These should be described clearly.

3. The role of the SNP marker detected in this study has not been displayed clearly.

Author Response

Reviewer #1

n this manuscript, a SNP marker on 2H that accounted the highest diversity index (Fst) values between cultivars and landraces and between 17 cultivars and wild accessions was found. The results obtained using this marker suggest that outlier locus may influence phyllochron variation which underwent positive selection in barley. This manuscript is interesting. However, some minor revision is needed.

1. In the Result section, SPOP1, SPOP2, SPOP3 and SPOPs were mentioned. They should be introduced in detail in Materials scetion.

>A paragraph is included in the material section to describe the population differentiation in details.

2. The manuscript has detected the difference of the gene AK366024 between wild and cultivar barley. Whether did this gene of wild barley exists in the introgression line S42IL109? Were the trait variations between line S42IL109 and Scarlett caused by the SNP in this gene? These should be described clearly.

>Yes, introgression line S42IL109 carried wild barley allele of AK366024 gene. AK366024 is one putative candidate gene associated to phenotype, but there are additional candidate genes as S472IL109 carried bigger wild barley introgression at the outlier locus on 2H. The primary focus of this study was the detection of outlier locus and its association with trait variation. In the present version, we have addressed this issue in the discussion part.

3. The role of the SNP marker detected in this study has not been displayed clearly.

>The SNP marker identified in this represents the outlier locus on chromosome 2H which underwent directional selection. Our series of phenotypic evaluation showed that the outlier locus putatively control the phyllochron in barely. This association is mentioned in the results and discussion sections.

Reviewer 2 Report

The manuscript by Reinert et al. describes a vital research with the use of genome-wide outlier loci in barely genetic population as a useful genetic test to identify loci underwent positive selection in barely. The authors evaluated the population structure of 179 barley genotypes, revealing three major clustering groups (SPOP1, 2 and 3) based on K-value. SNP outlier analysis in the barely diversity set using Bayescan showed 5 significant SNP markers. Among which, SCI_RS-170235 exhibited highest significant and commonly detected between SPOP1-3 and SPOP1-2. In addition, this SNP is located in the Protein ID: BAJ97227.1 and most of the wild barely carried "thymine" at position 224p while its missing in cultivated one. Further, the authors used Introgression lines of of wild barely carrying this SNP outlier loci with Scarlett German culrivar. The comparison between S42IL109 and Scarlett revealed a significant difference in phyllochron which could be attributed to this outlier locus. The manuscript provided a thorough analysis of forward genetic in barely related to identification of new locus that underwent positive selection in barley.  Description of methods should be in more detail with detailed rationales for experimental design. In addition, some style and format need to be improved.

Added the affiliation number to Leon and Naz unless Journal style is different.

L19 PCR-based

L49, L71 ...etc. Check format of the reference order and style its not uniformed and need a precise check through all the manuscript.

L61 to L63 the definition of "forward genetics" and "reverse genetics" is not clear. For example, "forward genetics" approach aims to identify the genetic basis responsible for a phenotypic trait and mapped the genes assoictaed with this trait. While"reverse genetics" is to study the function of a particular gene by studying its effect on the phenotype due to change in the sequence.

L74 Provide the full name first mentioned.

L186-188 what is the watering system used, is it based on field capacity percentage or soil water potential..etc?

In the conclusion. L366-371 The authors mentioned that prolonged development is advantageous for development of more productive tillers, biomass and longer grain filling? these not tested in the current study and no results supporting this claim. SO authors should provide such data or reference to support the finding.

Author Response

Reviewer #2

The manuscript by Reinert et al. describes a vital research with the use of genome-wide outlier loci in barely genetic population as a useful genetic test to identify loci underwent positive selection in barely. The authors evaluated the population structure of 179 barley genotypes, revealing three major clustering groups (SPOP1, 2 and 3) based on K-value. SNP outlier analysis in the barely diversity set using Bayescan showed 5 significant SNP markers. Among which, SCI_RS-170235 exhibited highest significant and commonly detected between SPOP1-3 and SPOP1-2. In addition, this SNP is located in the Protein ID: BAJ97227.1 and most of the wild barely carried "thymine" at position 224p while its missing in cultivated one. Further, the authors used Introgression lines of of wild barely carrying this SNP outlier loci with Scarlett German culrivar. The comparison between S42IL109 and Scarlett revealed a significant difference in phyllochron which could be attributed to this outlier locus. The manuscript provided a thorough analysis of forward genetic in barely related to identification of new locus that underwent positive selection in barley.  Description of methods should be in more detail with detailed rationales for experimental design. In addition, some style and format need to be improved.

Added the affiliation number to Leon and Naz unless Journal style is different.

>Corrected

L19 PCR-based

>Corrected

L49, L71 ...etc. Check format of the reference order and style its not uniformed and need a precise check through all the manuscript.

>Reference format is corrected throughout the manuscript.

L61 to L63 the definition of "forward genetics" and "reverse genetics" is not clear. For example, "forward genetics" approach aims to identify the genetic basis responsible for a phenotypic trait and mapped the genes assoictaed with this trait. While"reverse genetics" is to study the function of a particular gene by studying its effect on the phenotype due to change in the sequence.

>We agree and it is improved as suggested.

L74 Provide the full name first mentioned.

>Corrected

L186-188 what is the watering system used, is it based on field capacity percentage or soil water potential..etc?

>The details of irrigation system and soil water potential are included in the material and method section. Soil water content was measured using volumetric moisture content.

In the conclusion. L366-371 The authors mentioned that prolonged development is advantageous for development of more productive tillers, biomass and longer grain filling? these not tested in the current study and no results supporting this claim. SO authors should provide such data or reference to support the finding.

>The statement is purely based on our hypothesis for a putative positive selection at the outlier locus as well as to present an outlook to conduct further research on this locus. We have improved this statement in the present version.